# GAUSSIANS ON A DIET: HIGH-QUALITY MEMORY-BOUNDED 3D GAUSSIAN SPLATTING TRAINING

## ABSTRACT

3D Gaussian Splatting (3DGS) has revolutionized novel view synthesis with high-quality rendering through continuous aggregations of millions of 3D Gaussian primitives. However, it suffers from a substantial memory footprint, particularly during training due to uncontrolled densification, posing a critical bottleneck for deployment on memory-constrained edge devices. While existing methods prune redundant Gaussians post-training, they fail to address the peak memory spikes caused by the abrupt growth of Gaussians early in the training process. To solve the training memory consumption problem, we propose a systematic memory-bounded training framework that dynamically optimizes Gaussians through iterative growth and pruning. In other words, the proposed framework alternates between incremental pruning of low-impact Gaussians and strategic growing of new primitives with an adaptive Gaussian compensation, maintaining a near-constant low memory usage while progressively refining rendering fidelity. We comprehensively evaluate the proposed training framework on various real-world datasets under strict memory constraints, showing significant improvements over existing state-of-the-art methods. Particularly, our proposed method practically enables memory-efficient 3DGS training on NVIDIA Jetson AGX Xavier, achieving similar visual quality with up to 80% lower peak training memory consumption than the original 3DGS. Our demo page is available at http://dietgaussian.work.gd/.

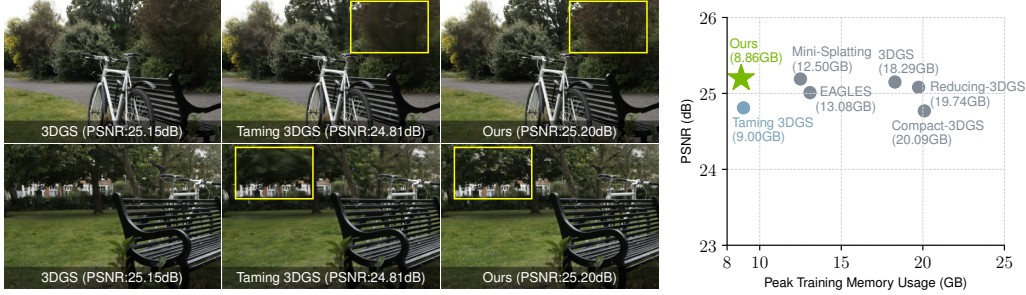

Figure 1: We present a memory-bounded 3D Gaussian Splatting training framework, enabling lower peak training memory and higher rendering quality, compared to existing state-of-the-art methods.

## 1 INTRODUCTION

3D Gaussian Splatting (3DGS) (Kerbl et al., 2023) has recently emerged as a powerful paradigm for novel view synthesis and 3D reconstruction (Chen & Wang, 2024). By representing a scene as a set of 3D Gaussians, each with parameters such as spatial position, scale, opacity, rotation, and spherical harmonic (SH) coefficients for view-dependent color. 3DGS enables differentiable rendering with promising visual quality. This approach has demonstrated state-of-the-art performance in rendering speed and quality, achieving immersive view synthesis at high resolutions in real time. However, these gains come at a substantial memory cost – 3DGS models often employ millions of Gaussians for a single scene, leading to significant memory consumption (Bagdasarian et al., 2024). This reliance on a large number of primitives not only inflates the model size but also restricts deployment on edge devices or other memory-constrained platforms (Morgenstern et al., 2024; Yu

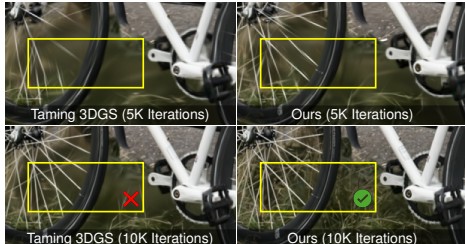

Figure 2: **Gaussian number vs. iteration.** Our method grows and prunes Gaussians under memory constraints, while 3DGS and Mini-Splatting densify to millions and remove later.

Figure 3: **Convergence comparison.** Taming 3DGS fails to fix errors (e.g., grass), while our method continues optimizing those areas.

et al., 2024a). In practice, the heavy memory footprint of 3DGS-based models has become a key bottleneck, limiting their scalability and adoption in resource-limited settings.

Existing works (Deng et al., 2024; Fang & Wang, 2024; Niemeyer et al., 2024; Papantonakis et al., 2024; Zhang et al., 2024b) mainly focus on pruning redundant Gaussians to obtain a compact scene representation. Nevertheless, the challenging problem is that existing pruning approaches are applied after the uncontrollable densification process in the original 3DGS training framework (Fan et al., 2023; Rota Bulò et al., 2024; Kheradmand et al., 2024; Yu et al., 2024b; Lee et al., 2024a), where the Gaussian primitives suddenly expand to a tremendous number as shown in Fig. 2, e.g., several million for the bycicle scene, leading to a substantial peak training memory consumption (refer to Fig. 1[1]). Even though those methods successfully reduce the memory footprints in the rendering phase, the peak memory consumption is significantly higher than the memory size of edge systems, thereby hindering real-time 3D applications in real-world settings (Matsuki et al., 2024).

Despite the practical significance of peak training memory usage in 3DGS, this issue remains understudied. Prior work (Mallick et al., 2024) mitigates memory spikes by regulating Gaussian growth via a predictable curve and selectively cloning/splitting primitives using a computationally intensive importance score. While this approach reduces peak Gaussian counts, it suffers from some key drawbacks. (1) Strict growth restrictions in early training stages limit representational capacity, leading to accumulated rendering errors that persist due to insufficient dynamic refinement. (2) Cloned Gaussians inherit identical initial positions, resulting in redundant gradient updates that fail to effectively capture missing scene details. (3) The absence of an active Gaussian removal mechanism allows poorly optimized ("ill") Gaussians to persist throughout training.

To address those limitations and practically enable real-time 3DGS training on memory-constrained devices, in this paper, we conduct an in-depth study on the unsatisfactory performance of existing training approaches. Our motivation is inspired by the Lottery Ticket Hypothesis (Frankle & Carbin, 2018),which suggests the existence of a sparse subnetwork capable of achieving performance comparable to a dense one. We extend this hypothesis to 3DGS, whether an optimal sparse Gaussian model can be trained from scratch under strict memory limitations. Building on this insight, we propose a systematic memory-bounded 3DGS training framework based on dynamic growing and removal of Gaussian primitives, which can strictly satisfy the practical memory constraints. Our proposed training framework alternatively identifies, grows, and prunes Gaussian primitives in each iteration, where "ill" Gaussians are dynamically deleted and "healthy" Gaussians are subsequently regenerated. This iterative process ensures consistently low memory usage while discovering effective primitives that match the rendering quality of the original 3DGS at significantly reduced training memory consumption. As illustrated in Fig. 1, our dynamic strategy outperforms one-shot pruning (Fang & Wang, 2024; Girish et al., 2024) in both memory efficiency and visual fidelity.

Specifically, we first densify the Gaussians to the user-specified bound in the early training stage, and then we iteratively grow and prune a small proportion of Gaussians until the training ends. Our approach iteratively alternates between three steps: (1) Dynamical growing – we introduce a clone-

---

[1]Here, we report our peak memory usage on GTX 4090 without engineering optimization for a fair comparison, while Mini-Splatting (Fang & Wang, 2024) leverages extra compression by downgrading the orders of SH coefficients to one before pruning to reduce memory.

and-split strategy based on the hybrid position and color gradients to coarse-grainly refine rendering quality. (2) Adaptive Gaussian compensation – we propose a mechanism that fine-grainly adds new Gaussian primitives to the identified underfitting regions. By projecting high-reconstruction-error pixels back to 3D space, we generate Gaussians in poorly represented areas, progressively refining detail. (3) Memory-aware pruning – to maintain a fixed memory budget, we remove low-importance Gaussians whenever the total count exceeds a predefined threshold based on the light-computation criterion. This ensures balanced growth while preserving critical scene structures. By iteratively refining the Gaussian set, adding discriminative primitives, and removing redundant ones, our framework constructs a compact yet expressive representation that matches the original 3DGS in quality while drastically reducing memory overhead. In summary, our contributions are as follows:

- We conduct an in-depth study on the inefficiency and inferior quality of existing 3DGS training methods.

- We develop a systematic memory-efficient training framework that dynamically optimizes Gaussians via iterative growth and pruning, achieving high rendering quality under strict memory bounds.

- We propose a dual-level Gaussian growing approach, i.e., coarse-grained clone/split based on hybrid gradients, fine-grained compensation based on pixel-level rendering errors, and a dynamic primitive shift strategy reducing ineffective gradients update.

- We comprehensively evaluate our proposed training framework on various real-world datasets, achieving improved rendering quality and reduced peak training memory. Particularly, we practically test on-device training on NVIDIA Jetson AGX Xavier, providing up to 80% peak memory reduction with similar quality compared to the original 3DGS.

## 2 RELATED WORK

### 2.1 COMPACT 3D GAUSSIAN SPLATTING

Although 3DGS achieves significant progress in photorealistic scene representation and novel view synthesis, its reliance on millions of primitives creates significant memory bottlenecks that hinder practical deployment (Bagdasarian et al., 2024; Hanson et al., 2024; Lee et al., 2025). This challenge has led to growing interest in compact 3DGS methods, which aim to preserve rendering fidelity while significantly reducing the number of primitives (Bao et al., 2025; Liu et al., 2024; Ye et al., 2024). For instance, LightGaussian (Fan et al., 2023) reduces final storage by pruning redundant Gaussians based on a global importance score after training, while RadSplat (Niemeyer et al., 2024) improves pruning robustness by replacing the sum with a max operator for score computation.

Although these post-training pruning strategies reduce memory usage during inference, they do not alleviate the high peak memory consumption incurred during training (Feng et al., 2024; Navaneet et al., 2024; Zhang et al., 2024a). To address this, Taming 3DGS (Mallick et al., 2024) introduces a steerable densification mechanism that selectively densifies impactful Gaussians, enabling a more predictable and memory-aware growth trajectory. However, its infrequent densification and slow growing speed result in suboptimal and poorly positioned Gaussians remaining in the model, limiting the representation details and overall rendering quality (see Fig. 3). Based on the in-depth study on the limitations of (Mallick et al., 2024), our work progressively refines the model via iteratively growing and pruning, dynamically preserving most "healthy" Gaussians under memory bounds.

## 3 BACKGROUND AND MOTIVATION

### 3.1 BACKGROUND OF 3D GAUSSIAN SPLATTING

3D Gaussian Splatting (3DGS) (Kerbl et al., 2023) represents the scenes using an optimized collection of anisotropic 3D Gaussians. Each Gaussian $G$ is defined by its covariance matrix $\boldsymbol{\Sigma}$ and center position $\boldsymbol{\mu}$ as

$$G(\boldsymbol{x}) = \exp\left(-\frac{1}{2}(\boldsymbol{x} - \boldsymbol{\mu})^\top \boldsymbol{\Sigma}^{-1}(\boldsymbol{x} - \boldsymbol{\mu})\right),$$

(1)

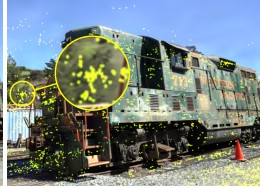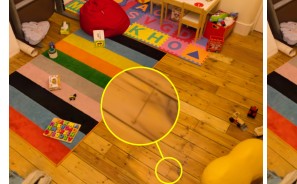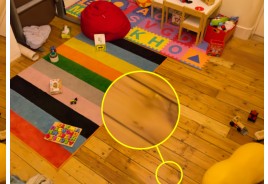

Figure 4: **Illustration of Gaussian compensation.** (Left) Color gradient per pixel. (Right) Compensated Gaussians in yellow color. Our compensation step recovers the high-frequency region that is hard to capture by the original densification (e.g., rubble under the train).

Figure 5: **Rendered images with our growing strategy.** (Left) Our proposed hybrid gradient-based method recovers the texture of the floor more accurately. (Right) Existing approaches based on position-only gradient lose details.

where $\boldsymbol{x}$ is an arbitrary position in the 3D scene. The covariance matrix $\boldsymbol{\Sigma}$ is generally decomposed as $\boldsymbol{\Sigma} = \boldsymbol{R}\boldsymbol{S}\boldsymbol{S}^\top\boldsymbol{R}^\top$, where $\boldsymbol{R}$ is a rotation matrix and $\boldsymbol{S}$ is a diagonal scaling matrix.

To render a 2D image from the 3D scene, 3DGS projects 3D Gaussians onto the image plane based on the camera parameters. The projected 2D covariance matrix is computed as $\boldsymbol{\Sigma}' = \boldsymbol{J}\boldsymbol{W}\boldsymbol{\Sigma}\boldsymbol{W}^\top\boldsymbol{J}^\top$ where $\boldsymbol{W}$ represents the view transformation matrix, and $\boldsymbol{J}$ is the Jacobian of the affine approximation of the projective transformation. Then, the final color $C$ at each image pixel is computed by blending all $N$ depth-ordered Gaussians contributing to the pixel as

$$C = \sum_{i \in N} c_i \alpha_i \prod_{j=1}^{i-1}(1 - \alpha_j). \qquad (2)$$

Here, $c_i$ is the color of each Gaussian derived from the SH coefficients. $\alpha_i$ is the ray transmittance calculated by overlapped Gaussians' opacity $o_i$ and the relative distance between rendered pixel position $x$ and 2D view-plane Gaussian's center $\mu_i$ (Niemeyer et al., 2024), i.e.,

$$\alpha_i(\boldsymbol{x}) = o_i \exp\left(-\frac{1}{2}\left(\boldsymbol{x} - \mathcal{R}\left(\boldsymbol{\mu}_i; \theta\right)\right)^\top \mathcal{R}_\theta\left(\boldsymbol{\Sigma}_i\right)^{-1}\left(\boldsymbol{x} - \mathcal{R}\left(\boldsymbol{\mu}_i; \theta\right)\right)\right), \qquad (3)$$

where $\theta$ is the camera pose and $\mathcal{R}$ is the 3D-to-image plane projection operation.

During training, the Gaussians are initialized from a sparse point cloud generated by Structure-from-Motion (SfM) (Schonberger & Frahm, 2016). Then, each Gaussian attribute is optimized with the gradient backpropagation to minimize the reconstruction error.

### 3.2 MOTIVATION FOR RESOLVING PROBLEMS IN 3DGS TRAINING

We conduct an in-depth study on existing training approaches, including the original 3DGS (Kerbl et al., 2023) and subsequent pioneering works (Mallick et al., 2024), which propose the accelerated 3DGS training. Our analysis highlights significant limitations in these methods, motivating the development of a new training framework that dynamically grows and removes Gaussians based on more effective criteria. Below, we present three main explorations of existing approaches.

*① Existing strategies cannot adjust Gaussian primitives dynamically, leading to accumulated error.* The densification process in 3DGS (Kerbl et al., 2023) is governed by the adaptive density control, which operates on a predetermined schedule. During the densification, the positional gradient magnitude for each Gaussian is tracked and averaged over all rendered views, resulting in a score. If it exceeds a user-defined threshold, the primitive is considered for growth through either cloning or splitting, depending on its size as determined by the scaling matrix. To solve the uncontrollable number of Gaussians and the challenges in threshold determination, the subsequent methods (Mallick et al., 2024; Rota Bulò et al., 2024) design a parabolic curve to define a schedule of new primitives at each step. Based on the predictable densification curves, they add new Gaussians by cloning or splitting existing ones according to the developed complex importance and error-correction scores.

Even though the above methods can effectively regulate the number of Gaussians, they grow the Gaussians slowly and achieve the user-specified budget after a long-term period, i.e., 15,000 iterations. This growing strategy limits the representation power due to the limited number of Gaussians

before reaching the budget, leading to a performance drop. Additionally, existing strategies prune unimportant Gaussians based on opacity (Lu et al., 2024) or other criteria (Lee et al., 2024b) in every long-term iteration, e.g., 500 iterations, which fails to figure out the truly important Gaussians. This lies in that after 500 iterations, the "error" information is mixed among all Gaussians, making it challenging to remove the redundancy without frequent operation on those Gaussians. As shown in Fig. 3, with the two main limitations, existing methods cannot correctly optimize Gaussians under a bounded number of Gaussians.

**Proposed solution:** Inspired by the sparse training method (Frankle & Carbin, 2018; Han et al., 2016) for under-parameterized neural networks, we propose a dynamic approach that grows and removes Gaussians frequently (e.g., every 50 iterations) and adaptively. In our proposed training framework, at every step, we recognize a small proportion of "ill" Gaussians based on the blending weight (which will be introduced in the next Section) instead of opacity. Then, after removing those "ill" Gaussians, we add the same size of new Gaussians in the needed area. With this adaptive strategy, we can keep a sufficient capacity of Gaussians at the beginning without crafted growing curves. Our results show that our dynamic method can find the "healthy" Gaussians and remove the redundancy effectively, outperforming existing work (Mallick et al., 2024) with fewer iterations.

②  *Naive clone or split limits the representation.* Following the 3DGS densification, prior methods (Kerbl et al., 2023; Zhou et al., 2024) clone or split Gaussian primitives according to positional gradients, and the added Gaussians overlap the original ones with similar parameters. This naive operation causes cloned Gaussians to receive similar gradient updates during optimization. The similarity in gradients hinders their ability to diverge spatially, leading them to remain overlapped for extended periods (Deng et al., 2024). Consequently, a significant number of low-opacity Gaussians persist in the scene, leading to redundancy that is challenging to mitigate while increasing computational and memory overhead. Besides, due to the lack of pixel-wise information, existing methods cannot deal with areas where more Gaussians are needed but gradients fail to recognize, leading to permanently low-quality rendering results.

**Proposed solution:** To resolve this issue, the densification process needs additional information to provide randomness, but the introduced information should have a negligible impact on the rendering results. In our proposed framework, we shift the added Gaussians by a small distance based on the accumulated positional gradient, reducing the overlap between the two primitives. In this way, those Gaussians can receive gradually distinct gradient updates, thereby the new Gaussians will move to the correct location adaptively, which benefits the rendering quality. In other words, our proposed solution strengthens the robustness of the scene representation. Additionally, we propose a pixel-wise Gaussian compensation method performed at the end of each step. Specifically, we select pixels with the highest losses and put extra Gaussians on the rays rendering those pixels. As shown in Fig. 4, our results show that our compensation identifies and recovers the high-frequency region that is hard to be captured by the original growth method.

③  *Growing by position-only gradient cannot capture the blurry areas that need more Gaussians.* 3DGS (Kerbl et al., 2023) initializes the scene using a sparse point cloud generated from Structure-from-Motion (SfM) (Schonberger & Frahm, 2016; Ullman, 1979), assigning default values to each Gaussian's attributes. Then, it employs an adaptive density control algorithm to add new Gaussians during the densification step. In the densification, if the Gaussians' view-space positional gradients exceed a predefined threshold, they are candidates for duplication. Specifically, Gaussians with scales above a certain threshold are split, and otherwise are cloned. This strategy uses the view-space position gradient, computed via per-pixel color gradients, as an indicator for duplication. Referring to Equation 2 and 3, we apply the chain rule to derive the gradient with respect to position $p_k$ of the $k$-th Gaussian, i.e.,

$$\frac{d\ell}{dp_k} = \frac{d\ell}{dC}\frac{dC}{d\alpha_k}\frac{d\alpha_k}{dp_k}, \tag{4}$$

where $\ell$ is the rendering loss. Then, we analyze the second term, the partial derivative of $C$ with respect to $\alpha_k$, and expand it with other variables based on Equation 2. To be specific, this term can be represented by $\frac{dC}{d\alpha_k} = \sum_{j=k}^{N} -c_k\alpha_k\Pi(1-\alpha_k)$. This formulation shows that position gradients only capture partial color information and are influenced by the magnitude of the overlapped Gaussians' color. As a result, position gradients often fail to accurately detect underfit areas, particularly in blurred or low-contrast regions.

Figure 6: **Overall workflow of our proposed memory-bounded 3DGS training framework,** which iteratively performs growing, compensation, and pruning, progressively refining the representation capability.

**Proposed solution:** According to Equation 4, we compute the gradient with respect to color $c_k$ of the $k$-th Gaussians, i.e.,

$$\frac{d\ell}{dc_k} = \frac{d\ell}{dC}\frac{dC}{dc_k}. \tag{5}$$

We also analyze the second term by expanding it as $\frac{dC}{dc_k} = \alpha_k \Pi_{j=1}^{k-1}(1-\alpha_j)$. It is seen that the gradient of color is not influenced by the other Gaussians' color and is only related to the transmittance. To address the aforementioned problem, we proposed to combine the two types of gradients, i.e., a mixture of both position and color gradients, to determine the areas where new Gaussians need to be added. Our investigation shows that our solution can accurately locate the blurry regions, allocating more primitives to complement the rendering quality, as shown in Fig. 5.

## 4 THE PROPOSED SYSTEMATIC TRAINING FRAMEWORK

### 4.1 FRAMEWORK OVERVIEW

In the previous section, we have in-depth analyzed the limitations of existing 3DGS training methods, and we have introduced the motivations for our proposed framework. In this section, we will present our memory-efficient and high-quality training framework in detail. The overview of our proposed framework is illustrated in Fig. 6. In summary, our framework dynamically grows, compensates, and prunes Gaussians in an iterative way, where those steps are alternately performed in each iteration, progressively refining the representative capability under a consistent memory bound. In the growing step, we clone and split Gaussians based on the proposed hybrid gradients criterion at a coarse-grained level. In other words, this step rapidly increases the number of Gaussians during the early densification, then our method continuously introduces new Gaussians at the rest of the clone step to refine the model by compensating for the representation capability in the low-quality areas. In the Gaussian compensation step, we fine-grainly identify the low-quality pixels with the highest error, project them back to their corresponding 3D location, and generate new Gaussians at that point to better capture underfitting regions. On the other hand, in the pruning phase, to ensure the model remains within a memory budget, we concurrently remove an equal number of less important Gaussians when the total count exceeds a predefined threshold. By iteratively performing these two steps, our framework adaptively discovers and optimizes a compact subset of Gaussians that preserves rendering fidelity while ensuring memory efficiency throughout training. The overall algorithm is presented in Algorithm 1.

### 4.2 ITERATIVE GROWING AND PRUNING

The principle of our training framework is to alternately operate growing and pruning steps in each iteration, dynamically adding informative Gaussians and removing redundant Gaussians. For the growing step, as discussed in section 3.2, we incorporate the color gradient, $\nabla_c$, which reflects rendering error more accurately in the actual rendering space. According to the gradient backpropagation formulation, Eq. 4 and Eq. 5, color error flows entirely into the color gradients of individual Gaussians after being scaled by the transmittance weight. Therefore, we propose a mixed criterion

that leverages color and position information to better identify regions requiring densification, i.e., $\nabla_{\mathrm{mix}} = \nabla_{\mathrm{p}} + \nabla_{\mathrm{c}}$, to clone and split Gaussians.

As the comparison shown in Fig. 4, color gradients more reliably highlight underfitting regions, particularly in backgrounds and blurred areas. Our proposed hybrid metric provides a more informative cue for identifying and densifying under-structured geometric regions, as shown in Fig. 5.

**Dynamical position adjustment.** To address the overlapping problem in existing densification approaches that directly copy Gaussians, we propose an adaptive position adjustment method based on the accumulated gradient information to move new Gaussians to appropriate places. Specifically, we accumulate all position gradients over $N$ views, i.e.,

$$\mu_{\mathrm{new}} = \mu_{\mathrm{old}} + \sum_{i \in N} \nabla_{\mu i} \tag{6}$$

representing a stable and optimal position where new Gaussians should be. This light shift effectively resolves the overlapping problem that new Gaussians receive similar updating gradients.

In each iteration, as new Gaussians are added in the previous growing steps, an equal number of the least important Gaussians are subsequently removed in this pruning step, keeping the total peak training memory under the constraint. To achieve iterative pruning, it requires continuously identifying and removing the least important Gaussians. However, calculating a comprehensive importance score for each Gaussian can be computationally expensive. We apply the importance criterion (Niemeyer et al., 2024) with light calculation by aggregating the ray contribution of Gaussians $i$ along all rays of $N$ views. For Gaussians $G_i$,

$$R_i = \max_{r \in R_f} \alpha_i^r \tau_i^r \tag{7}$$

where $R_f$ represent all rays in the $N$ views, and $\tau_i = \alpha_i \prod_{j=1}^{i-1}(1 - \alpha_j)$. This ray-based metric reflects the blending contribution of each Gaussian to the final pixel color and can be computed efficiently within the existing rendering pipeline, avoiding any significant overhead.

In summary, our iterative growing and pruning have two advantages: firstly, it enables consistent training on devices with strict memory constraints where one-shot pruning approaches (Fan et al., 2023; Fang & Wang, 2024) fail. Secondly, it allows the model to recover and re-optimize after each pruning, leading to a more balanced and high-quality sparse representation.

### 4.3 ADAPTIVE GAUSSIAN COMPENSATION

Positions of Gaussians receive only infinitesimal fluctuating gradients, updated by gradients propagated through the chain rule across varying camera views. As training progresses, the exponentially decaying learning rate further leaves the Gaussian stable, while reconstruction errors in underfit, blurry regions persist. To refine the poorly reconstructed and sparsely covered areas, we innovatively propose a Gaussian compensation approach before the pruning step to generate new Gaussians in the underfitting area based on a per-pixel error that measures the difference between the ground truth and the rendered image. Per-pixel error can be directly derived from the color gradient for each pixel, computed in the original backward pass, without incurring additional computational cost.

Once underfitting pixels are identified, the next challenge is transforming 2D pixel coordinates into corresponding 3D Gaussian positions. We can replace the color $c_i$ of the $i$-th Gaussian with the depth of its center $d_i$ as

$$D = \sum_{i \in N} d_i \alpha_i \prod_{j=1}^{i-1}(1 - \alpha_j). \tag{8}$$

This approach approximately estimates the depth for each pixel. Then, inspired by (Fang & Wang, 2024), we project the selected high-error pixels back into 3D space by replacing $d_i$ with $d_{\mathrm{mid}}$, the Gaussian midpoints that contribute most to the pixel. For each image, we identify the top-$K$ pixels with the highest color gradient magnitude and generate new Gaussians at the corresponding positions derived from $\alpha$-blended depth. We set the colors of these Gaussians same as the ground truth pixel values.

To be specific, as shown in Algorithm 2, our Gaussian compensation identifies the top-$K$ pixels $\boldsymbol{p}_{\mathrm{err}}$ with the highest errors and computes their corresponding 3D positions $\boldsymbol{D}_{\mathrm{err}}$ in world space using

**Algorithm 1** The overall procedure of our training framework.

**Input:** Gaussian primitives $G$, target peak number of Gaussians $F$, maximum number of iterations $T$;

1: $t \leftarrow 1$;
2: **while** $t < T$ **do**
3:     **if** densifyBegin $< t <$ densifyEnd **then**
4:         **if** $|G| < F$ **then**
5:             cloneAndSplit($G$);
6:             shiftNewGaussians($G$);
7:         **end if**
8:         prune($G, o_i < o_t$);
9:     **else if** compensateBegin $< t <$ compensateEnd **then**
10:         compensateGaussians($G$);
11:     **end if**
12:     **if** $|G| > F$ **then**
13:         $K = F - |G|$;
14:         $G' \leftarrow$ findLeastK($G, K$);
15:         prune($G, G'$);
16:     **end if**
17:     $t = t + 1$;
18: **end while**

**Algorithm 2** The proposed Gaussian compensation.

**Input:** Camera view $v$, ground truth image $I_{gt}$

1: $I \leftarrow$ render($v$)
2: $p_{err}, p_{gt} \leftarrow$ findErrorPixels($I, I_{gt}$)
3: $D_{err} \leftarrow$ renderDepth($p_{err}$)
4: list.append($D_{err}, p_{gt}$)
5: **if** $t$ % compensateInterval $= 0$ **then**
6:     genGaussians(list)
7:     list.clear()
8: **end if**

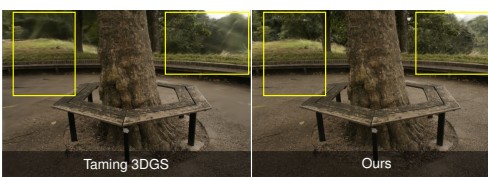

Figure 7: **Rendered image example.** Our method presents significantly higher perceptual quality with high-frequency details, while Taming 3DGS shows blurry background trees and land.

depth alpha-blending. The ground truth color at each selected pixel, denoted as $p_{gt}$, is also stored. After every user-defined interval, new Gaussians are generated at the computed positions $D_{err}$, and each is assigned the corresponding color $p_{gt}$. Fig. 4 shows that the compensated Gaussians in our proposed approach recover the underfitting regions missed by the original 3DGS, improving reconstruction quality in blurry or sparse areas.

Table 1: **Quantitative results on multiple datasets, compared with existing state-of-the-art works.** Reducing-3DGS, Compact-3DGS and EAGLES results are replicated using official code. 3DGS, Mini-Splatting and Taming 3DGS results are reported from (Mallick et al., 2024). "#G/M" denotes the **peak** number of Gaussians in training (in millions). The **absolutely best results** are shown in bold, and the best results from efficient training methods are highlighted. Horizontal bars provide an intuitive comparison of the peak number of Gaussian points. "↓" and "↑" indicate lower and higher values are better, respectively.

| Method | Mip-NeRF 360 | | | | Tanks&Temples | | | | Deep Blending | | | |
|---|---|---|---|---|---|---|---|---|---|---|---|---|
| | PSNR↑ | SSIM↑ | LPIPS↓ | #G/M↓ | PSNR↑ | SSIM↑ | LPIPS↓ | #G/M↓ | PSNR↑ | SSIM↑ | LPIPS↓ | #G/M↓ |
| 3DGS (Kerbl et al., 2023) | 27.46 | 0.815 | 0.215 | 3.310 | 23.65 | 0.847 | 0.176 | 1.840 | 29.64 | 0.904 | 0.243 | 2.810 |
| Mini-Splatting (Fang & Wang, 2024) | 27.26 | **0.822** | **0.217** | 4.320 | 23.42 | **0.847** | **0.181** | 4.320 | **30.04** | **0.910** | **0.244** | 4.510 |
| Reducing-3DGS (Papantonakis et al., 2024) | 27.21 | 0.811 | 0.225 | 2.749 | 23.59 | 0.841 | 0.187 | 1.507 | 29.61 | 0.903 | 0.248 | 2.218 |
| Compact-3DGS (Lee et al., 2024b) | 26.96 | 0.797 | 0.244 | 2.590 | 23.34 | 0.831 | 0.202 | 1.465 | 29.80 | 0.900 | 0.257 | 2.268 |
| EAGLES (Girish et al., 2024) | 27.15 | 0.811 | 0.231 | 1.928 | 23.27 | 0.837 | 0.201 | 0.954 | 29.83 | 0.909 | 0.246 | 1.981 |
| Taming 3DGS (Mallick et al., 2024) | 27.22 | 0.795 | 0.260 | 0.632 | **23.68** | 0.836 | 0.211 | 0.319 | 29.49 | 0.900 | 0.270 | 0.294 |
| **Ours** | 27.30 | 0.809 | 0.234 | **0.628** | 23.62 | 0.842 | 0.192 | **0.318** | 29.64 | 0.906 | 0.256 | **0.292** |

## 5 EVALUATION

**Dataset and metrics.** Following the standard practice, we evaluate our rendering performance on three novel view synthesis datasets: Mip-NeRF 360 (Barron et al., 2022), Tank&Temple (Knapitsch et al., 2017), and Deep-Blending (Hedman et al., 2018). For quantitative evaluation, we report peak signal-to-noise ratio (PSNR), structural similarity (SSIM), and learned perceptual image patch similarity (LPIPS) (Zhang et al., 2018). Furthermore, we assess memory efficiency by measuring peak training memory usage on real-world edge settings, i.e., NVIDIA Jetson AGX Xavier. For

visualized evaluation, we show rendering results of 3DGS (Kerbl et al., 2023) and Taming 3DGS (Mallick et al., 2024) on various scenes for comparison.

**Implementation details.** All render quality experiments are conducted under the same environment specified in the original 3DGS (Kerbl et al., 2023) and Taming 3DGS (Mallick et al., 2024) using an NVIDIA GTX 4090 GPU. Following Mini-Splatting (Fang & Wang, 2024), we reset all Gaussians' opacity and position at the 5K-th iteration for Mip-NeRF 360 (Barron et al., 2022) outdoor scene. Our Gaussian compensation step starts at the 10K-th iteration and ends at the 15K-th iteration. After that, we fine-tune the result to a certain iteration depending on each scene.

## 5.1 QUANTITATIVE RESULTS

Quantitative results are summarized in Table 1, in comparison with the original 3DGS (Kerbl et al., 2023) and the state-of-the-art training method Taming 3DGS (Mallick et al., 2024). We also compare to the state-of-the-art pruning works like Mini-Splatting (Fang & Wang, 2024). It is seen that we outperform Taming 3DGS (Mallick et al., 2024) by an average of 0.15 dB PSNR and 0.03 LPIPS with fewer peak Gaussians across all scenes. Compared to the state-of-the-art pruning method, Mini-Splatting (Fang & Wang, 2024), and the vanilla 3DGS (Kerbl et al., 2023), our method improves PSNR by 0.5 dB on the Tank&Temple dataset and reduces peak numbers of Gaussians by more than $6\times$.

More importantly, our method practically achieves on-device training on memory-constrained platforms. Experiments conducted on Jetson Xavier reveal our method reduces peak memory usage by nearly $2\times$ compared to the original 3DGS, as shown in Table 2(b). Notably, we observed that up to three-quarters memory is used for dataset storage, we develop a parallel dataloader that dynamically prefetches and moves data between the storage and the memory according to the training pipeline. This effort further reduces peak training memory by more than 5 GB.

## 5.2 VISUAL QUALITY RESULTS

We compare the rendered images for the perceptual analysis, as illustrated in Fig. 7. It is seen that our method brings significantly higher visual quality with high-frequency and textural fidelity. On the contrary, Taming 3DGS (Mallick et al., 2024) loses details in textured regions such as tree bark and gravel surfaces. As noted in (Zhang et al., 2018), such smoothness can artificially inflate PSNR scores (e.g., Taming 3DGS: 26.48dB vs. Ours: 24.36dB), which explains why our method yields lower PSNR yet achieves better perceptual quality, as reflected by LPIPS. Fig. 8 and Fig. 9 show more rendered images on multiple scenes compared to Taming 3DGS (Mallick et al., 2024) and the original 3DGS (Kerbl et al., 2023). Our approach consistently achieves superior visual quality, particularly in high-frequency regions such as textured lawns and patterned blankets, where fine details are more faithfully preserved.

Additionally, we provide visualizations of Gaussian ellipsoids on the playroom scene to illustrate the spatial distribution of Gaussians Fig. 10. Our method significantly reduces redundancy by eliminating excessive overlap among Gaussians (middle image) and dynamically allocates a higher density of Gaussians to texture-rich regions (right image), effectively capturing complex scene content under memory-bounded.

Moreover, Fig. 11 demonstrates that our actual rendering quality is superior to Taming 3DGS (Mallick et al., 2024) at the iterations. Those results further show the effectiveness of our iterative growing and pruning over the existing slow densification based on prediction. Additional visual quality comparisons are seen in the Appendix.

## 6 CONCLUSION

We have presented a memory-efficient training framework for 3DGS that dynamically balances primitive growth and pruning under strict memory constraints. By iteratively refining Gaussians, coarse-grainly growing hybrid gradient varying areas, fine-grainly compensating underfitting regions while removing redundant ones, our approach achieves high-fidelity rendering with significantly reduced peak training memory consumption throughout training.

## REPRODUCIBILITY STATEMENT

We have made efforts to ensure that our results are reproducible. Our source codes, including the training script, dataset, and checkpoints, have been available on our GitHub repository..

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

Table 2: (a) LPIPS for the "Playroom" and "Drjohnson" scences. "I.P." denotes "Iterative Pruning", "G.C." denotes "Gaussians Compensation". (b) Peak training memory usage on NVIDIA Jetson Xavier.

| (a) Ablation study | | |
|---|---|---|
| Method | Playroom | Drjohnson |
| Ours | 0.259 | 0.253 |
| Baseline | 0.279 | 0.270 |
| +I.P. | 0.264 | 0.257 |
| +G.C. | 0.259 | 0.253 |

| (b) Memory usage (GB) | |
|---|---|
| Method | Mem. |
| 3DGS | 18.59 |
| Taming 3DGS | 10.01 |
| Ours | **8.55** |
| Ours w/loader | **2.98** |

# A  ADDITIONAL ABLATION STUDY

We conduct ablation studies to show the effectiveness of the components in our training framework. We test on the Deep-Blending (Hedman et al., 2018) dataset and report LPIPS scores to quantify the contribution of each component. We stop densification after Gaussians exceed a target number in the original 3DGS (Kerbl et al., 2023) and report it as the baseline. Note that all configurations yield the same final number of Gaussians.

Our first contribution involves an iterative pruning strategy, including hybrid gradients to grow new Gaussians and position adjustment after cloning the Gaussians. This procedure enables continual model refinement and yields an improvement of approximately 0.015 in LPIPS. Subsequently, we introduce the proposed Gaussian compensation, as illustrated in Fig. 4, which results in a further LPIPS improvement of 0.005. This demonstrates that generating new Gaussians in the highest error pixel enhances perceptual fidelity in underfitting regions.

We also conducted an ablation study to evaluate the impact of using mixed gradients. The mixed gradient strategy yields a modest improvement on the test dataset (+0.04 dB PSNR) but shows a significant gain in training dataset (+0.87 dB PSNR) on the kitchen, indicating better realistic practical application.

# B  ADDITIONAL QUANTITATIVE RESULTS

We summarize additional quantitative results on the Mip-NeRF 360, Tanks&Temples, and Deep Blending datasets in Table 4, Table 5, and Table 3.

Table 3: Deep Blending per scene results. 3DGS results are reported from (Girish et al., 2024). Taming 3DGS (Mallick et al., 2024) results are replicated using official code.

| Scene | Method | PSNR↑ | SSIM↑ | LPIPS↓ | #G/M↓ |
|---|---|---|---|---|---|
| Drjohnson | 3DGS | 28.77 | 0.900 | 0.250 | 3.260 |
| | Taming 3DGS | **29.40** | 0.903 | 0.266 | 0.404 |
| | Ours | 29.33 | **0.904** | **0.253** | **0.400** |
| Playroom | 3DGS | 30.07 | 0.900 | 0.250 | 2.290 |
| | Taming 3DGS | 29.59 | 0.898 | 0.274 | **0.185** |
| | Ours | **30.04** | **0.908** | **0.259** | 0.185 |
| Average | 3DGS | 29.42 | 0.900 | 0.250 | 2.780 |
| | Taming 3DGS | 29.49 | 0.900 | 0.270 | 0.294 |
| | Ours | **29.69** | **0.906** | **0.256** | **0.292** |

Table 4: Mip-NeRF 360 per scene results. 3DGS results are reported from (Girish et al., 2024). Taming 3DGS (Mallick et al., 2024) results are replicated using official code.

| Scene | Method | PSNR ↑ | SSIM ↑ | LPIPS ↓ | #G/M↓ |
|---|---|---|---|---|---|
| Bicycle | 3DGS | 25.13 | 0.750 | 0.240 | 5.310 |
| | Taming 3DGS | 24.85 | 0.718 | 0.295 | 0.813 |
| | Ours | **25.20** | **0.759** | **0.244** | **0.800** |
| Bonsai | 3DGS | 32.19 | 0.950 | 0.180 | 1.250 |
| | Taming 3DGS | 31.86 | 0.936 | 0.220 | 0.413 |
| | Ours | **31.88** | **0.938** | **0.212** | **0.410** |
| Counter | 3DGS | 29.11 | 0.910 | 0.180 | 1.170 |
| | Taming 3DGS | 28.59 | 0.898 | 0.223 | 0.311 |
| | Ours | **28.77** | **0.901** | **0.214** | **0.310** |
| Flowers | 3DGS | 21.37 | 0.590 | 0.360 | 3.470 |
| | Taming 3DGS | 21.07 | 0.554 | 0.407 | 0.575 |
| | Ours | **21.16** | **0.590** | **0.354** | **0.570** |
| Garden | 3DGS | 27.32 | 0.860 | 0.120 | 5.690 |
| | Taming 3DGS | **27.43** | 0.858 | 0.126 | **1.900** |
| | Ours | 27.36 | **0.865** | **0.107** | **1.900** |
| Kitchen | 3DGS | 31.53 | 0.930 | 0.120 | 1.770 |
| | Taming 3DGS | 30.95 | 0.922 | 0.141 | 0.482 |
| | Ours | **31.35** | **0.924** | **0.135** | **0.480** |
| Room | 3DGS | 31.59 | 0.920 | 0.200 | 1.500 |
| | Taming 3DGS | **31.27** | 0.908 | 0.250 | 0.225 |
| | Ours | 31.16 | **0.911** | **0.240** | **0.220** |
| Stump | 3DGS | 26.73 | 0.770 | 0.240 | 4.420 |
| | Taming 3DGS | 26.01 | 0.735 | 0.293 | **0.480** |
| | Ours | **26.37** | **0.754** | **0.268** | **0.480** |
| Treehill | 3DGS | 22.61 | 0.640 | 0.350 | 3.420 |
| | Taming 3DGS | **22.95** | 0.624 | 0.386 | 0.482 |
| | Ours | 22.48 | **0.635** | **0.329** | **0.480** |
| Average | 3DGS | 27.45 | 0.810 | 0.220 | 3.110 |
| | Taming 3DGS | 27.22 | 0.795 | 0.260 | 0.632 |
| | Ours | **27.30** | **0.809** | **0.234** | **0.628** |

Table 5: Tanks&Temples per scene results. 3DGS results are reported from (Girish et al., 2024). Taming 3DGS (Mallick et al., 2024) results are replicated using official code.

| Scene | Method | PSNR↑ | SSIM↑ | LPIPS↓ | #G/M↓ |
|---|---|---|---|---|---|
| Train | 3DGS | 21.94 | 0.810 | 0.200 | 1.110 |
| | Taming 3DGS | 22.14 | 0.804 | 0.237 | **0.365** |
| | Ours | **22.24** | **0.815** | **0.214** | **0.365** |
| Truck | 3DGS | 25.31 | 0.880 | 0.150 | 2.540 |
| | Taming 3DGS | 25.22 | 0.868 | 0.184 | 0.272 |
| | Ours | **25.00** | **0.869** | **0.170** | **0.270** |
| Average | 3DGS | 23.63 | 0.850 | 0.180 | 1.830 |
| | Taming 3DGS | 23.68 | 0.836 | 0.211 | 0.319 |
| | Ours | **23.62** | **.842** | **0.192** | **0.318** |

## C USE OF LARGE LANGUAGE MODELS (LLMs)

We used the large language models to assist with grammar polishing. The core research contributions were developed without relying on LLM.

## D ADDITIONAL VISUAL EXPERIMENTS

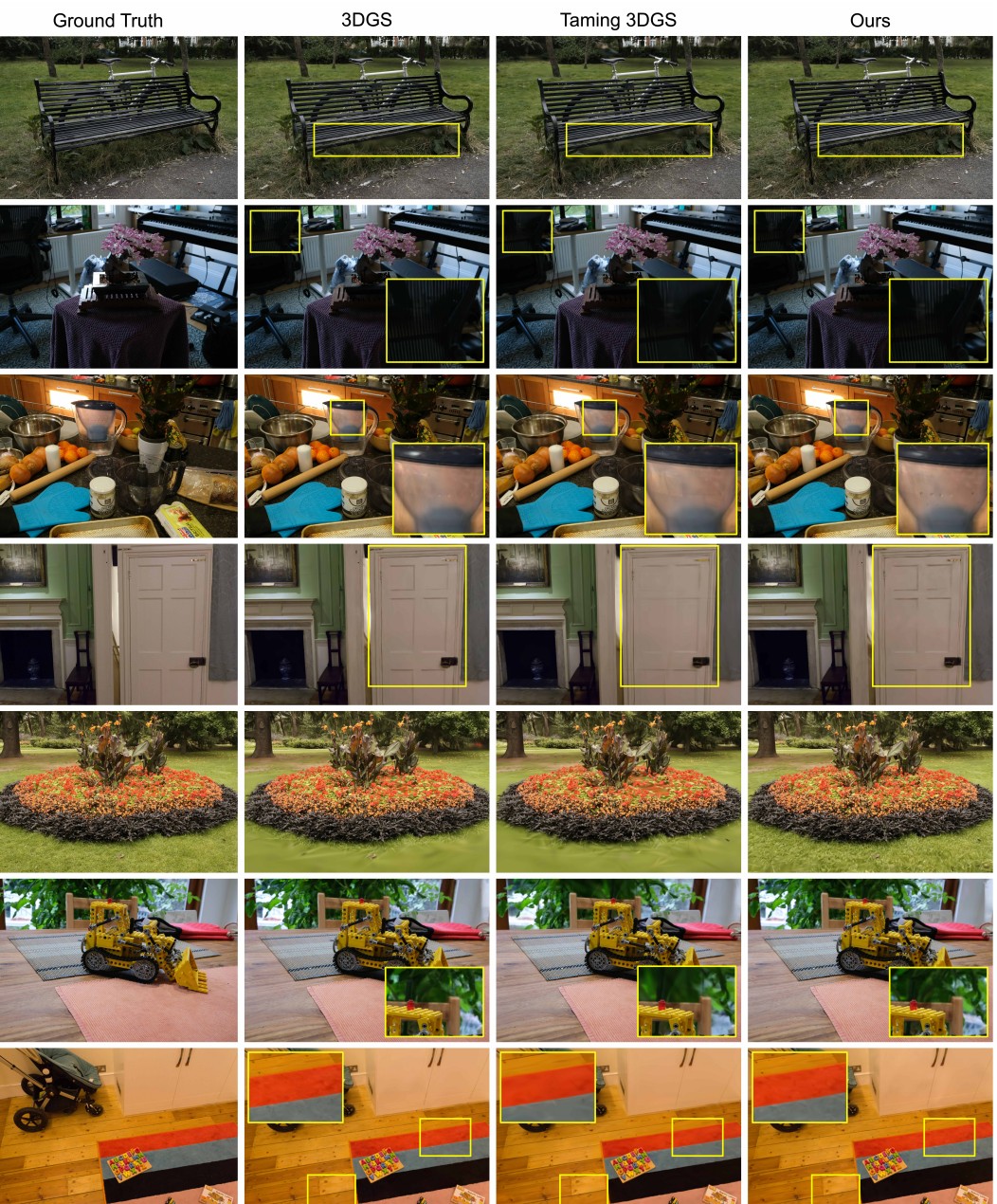

Figure 8: **Visualized results.** Our method achieves superior rendering quality compared against original 3DGS and Taming 3DGS (Mallick et al., 2024).

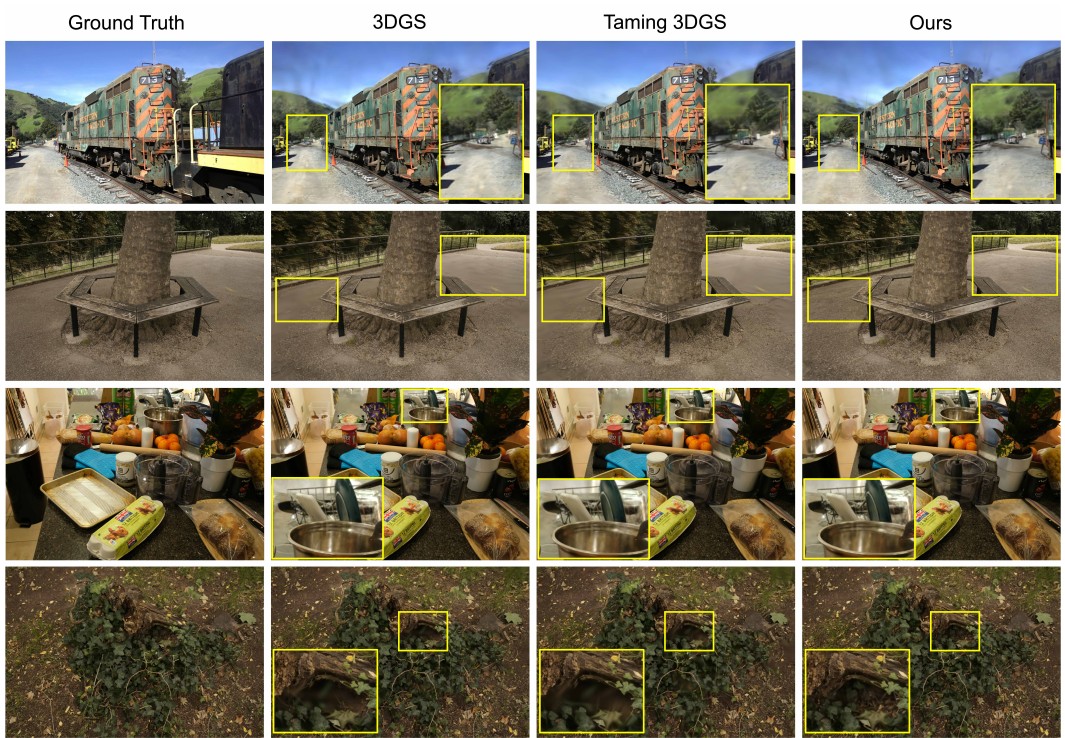

Figure 9: **Visualized results.** Our method achieves superior rendering quality compared against original 3DGS and Taming 3DGS (Mallick et al., 2024).

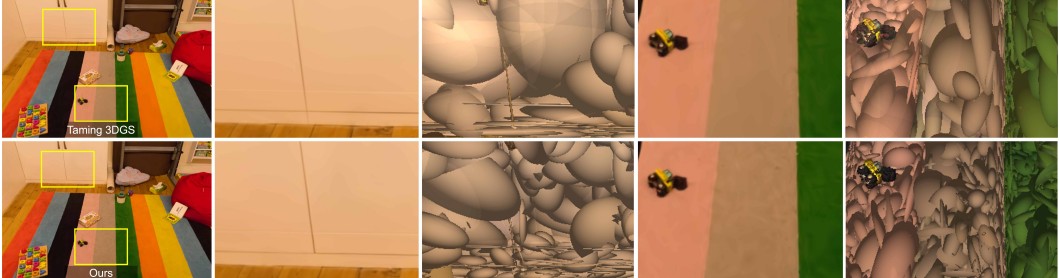

Figure 10: **Visualized ellipsoid results.** Our position adjustment reduces overlapped Gaussians (middle image), dynamically allocating more Gaussians to texture-rich regions (right image, texture of blanket), leading to a superior rendering quality.

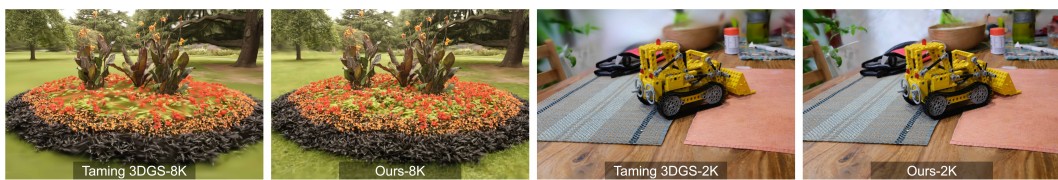

Figure 11: **Visualized results on flowers at the 8K-th iteration and kitchen at the 2K-th iteration.** Our method shows significantly improved rendering quality after the same training iterations compared to Taming 3DGS (Mallick et al., 2024).

