# OpenReview forum: "Gaussians on a Diet: High-Quality Memory-Bounded 3D Gaussian Splatting Training"
_ICLR.cc/2026/Conference — ICLR 2026 Conference Withdrawn Submission_

### Official Review · Reviewer_uUER · 2025-10-17

**Soundness:** 3
**Presentation:** 3
**Contribution:** 2
**Rating:** 4
**Confidence:** 3

**Summary:**

This paper tackles the high memory consumption of 3DGS during training. The authors propose a memory-bounded framework that alternates between iterative growing and pruning of Gaussian primitives to maintain a fixed memory budget. The method achieves comparable or superior rendering quality with lower peak memory than standard 3DGS, enabling on-device training on NVIDIA Jetson Xavier.

**Strengths:**

1.  This paper proposes a novel grow–prune–compensate strategy that dynamically controls the number of Gaussian primitives during training.
 2.  Experiments on standard datasets demonstrate both quantitative performance gains and significant memory reduction.

**Weaknesses:**

1.	The approach builds upon 3DGS with several engineering improvements. Although effective, the contribution seems incremental rather than fundamentally novel.
2.	How does the method’s training time compare to the baselines? Does the grow–prune strategy introduce additional overhead that offsets the memory savings?
3.	The ablation study is not sufficiently detailed, as it only reports LPIPS on two scenes and does not clearly show the contribution of each component. It would be helpful to further analyze the sensitivity to pruning frequency and memory budget, as well as the trade-off between memory constraints and reconstruction accuracy.

**Questions:**

The provided demo link in abstract appears to be inactive.

---

### Official Review · Reviewer_k2gG · 2025-10-23

**Soundness:** 2
**Presentation:** 3
**Contribution:** 2
**Rating:** 4
**Confidence:** 4

**Summary:**

The paper proposes “Gaussians on a Diet”, a memory-bounded training framework for 3D Gaussian Splatting (3DGS) that iteratively grows and prunes Gaussian primitives to maintain a near-constant low memory footprint during training. The core idea is to alternate between (1) dynamic growth using hybrid position–color gradients, (2) pixel-level Gaussian compensation in high-error regions, and (3) memory-aware pruning based on ray contribution.

**Strengths:**

1. Addresses an important practical issue: peak training memory in Gaussian Splatting.

2. Provides an organized overview of prior compact 3DGS methods and attempts to unify densification and pruning in one framework.

3. Demonstrates consistency across three datasets, suggesting a working implementation.

**Weaknesses:**

1. Most components (iterative pruning, hybrid gradients, pixel-error-based compensation) are straightforward adaptations of existing ideas.

2. Ablation analysis is incomplete and buried in the appendix. The main paper does not present any ablation results in the core sections, which severely limits the reader’s ability to assess the actual contribution of each proposed component (iterative pruning, hybrid gradients, Gaussian compensation). All relevant analyses only appear in the appendix without quantitative discussion or visual evidence in the main text. This omission significantly undermines the transparency and credibility of the claimed improvements.

**Questions:**

1. How long does your iterative growing-pruning framework take compared to Taming 3DGS? Is there additional computational overhead?

2. The memory usage on Jetson (Table 2b) shows “Ours w/loader 2.98 GB.” Does this include model parameters or just data loading? Please clarify the measurement methodology.

---

### Official Review · Reviewer_4xwR · 2025-10-29

**Soundness:** 4
**Presentation:** 4
**Contribution:** 3
**Rating:** 6
**Confidence:** 4

**Summary:**

This paper proposes a fine-grained Gaussian primitive control method for 3D Gaussian Splatting (3DGS). The approach dynamically adjusts the number of Gaussians at each iteration using a well-designed pruning and compensation algorithm to achieve improved performance. At the same time, it effectively satisfies memory constraints by maintaining a low peak memory footprint. Experimental results demonstrate that the proposed method achieves superior quantitative performance while using significantly fewer Gaussians compared to the baseline.

**Strengths:**

- This method dives deeply in the drawback of 3DGS, proposes a dynamic grow-and-prune strategy to maintain balanced capacity and efficient optimization.
- The proposed densification method introduces slight positional shifts and pixel-wise compensation to diversify gradients and enhance detail in high-error regions.
- By combining position and color gradients, the method accurately identifies blur regions and allocates additional primitives to improve rendering quality.
- The proposed method achieves superior performance under a lower memory budget, while maintaining a smaller model size without requiring post-training pruning.
- The paper also train 3DGS on embeded platform which is impressive.

**Weaknesses:**

- Although the number of primitives is reduced, frequent pruning and compensation may introduce extra computational overhead per iteration, which is not clearly quantified.
- The paper focuses heavily on empirical improvements but lacks a deeper theoretical justification for why frequent grow–prune operations stabilize training or improve convergence

**Questions:**

Since the main objective of this paper is to reduce training cost, I believe the authors should also report the training time. I am not suggesting that the method necessarily increases training time, but including this metric would make the evaluation more complete.

---

### Official Review · Reviewer_WB9M · 2025-11-01

**Soundness:** 2
**Presentation:** 2
**Contribution:** 3
**Rating:** 4
**Confidence:** 5

**Summary:**

This paper introduces several techniques to limit peak memory usage during the training of 3D Gaussian Splatting (3DGS) scenes. Existing methods run the densification algorithm at a fixed iteration frequency (e.g., every 100 in 3DGS), which inevitably slows convergence to a target number of Gaussians. In contrast, this paper proposes running the growing process (i.e., the densification algorithm) at every iteration, attaining target number of gaussians in a short time. To correctly identify under- and over-reconstructed Gaussians, the method additionally incorporates the color gradient into the densification criterion. To spatially diversify cloned Gaussians, it offsets their positions by the sum of positional gradients over N views, whereas existing methods place them at overlapping positions, causing the original and cloned Gaussians to receive coupled gradients. To refine poorly reconstructed and sparsely covered regions, the method also adds Gaussians in areas with high-error pixels at the median depth. Moreover, it prunes insignificant Gaussians identified by low maximum blending-weight scores at every iteration to control the total number of Gaussians. Experimental results show only minor quality degradation compared to vanilla 3DGS, but a significant reduction in the number of Gaussians, leading to lower peak memory usage suitable for edge devices such as the NVIDIA Jetson AGX Xavier.

**Strengths:**

This paper proposes a novel densification–pruning scheduling strategy that requires only a limited amount of peak memory, making it suitable for edge devices. Although some statements are arguable, the experiments demonstrate comparable reconstruction performance even with a highly constrained number of Gaussians.

**Weaknesses:**

This paper contains some debatable statements and an insufficient ablation study. I appreciate the impressive results, but additional evidence is needed to support the authors’ arguments. Please see Questions section.

**Questions:**

- My primary concern is training efficiency. How long does this method take to fully train (until 30k iterations)? I assume that since the growing, compensation, and pruning processes are executed at every iteration, the latency of a single iteration is significantly prolonged compared to vanilla 3DGS. Moreover, because this method’s fast-growing strategy ensures that the number of Gaussians quickly reaches the target in the early iterations, the rendering latency also start to surge in early iterations, leading to an overall longer total training time. I suggest that the authors include the measured training time in Table 1 to show training efficiency.

- In line 221, the authors state, "As shown in Fig. 3, with the two main limitations, existing methods cannot correctly optimize Gaussians under a bounded number of Gaussians." I do not think this statement is valid because Figure 3 actually compares rendered images at 5k and 10k iterations between Taming 3DGS and the proposed method, and according to the graph in Figure 2, these methods do not have the same number of Gaussians at the iterations. It appears that Taming 3DGS generates about one-third the number of Gaussians compared to the proposed method at 5k iterations. This fact is important since existing methods like Taming 3DGS still use long-periodic densification, their reconstruction quality at early iterations is expected to be poor. To support the authors’ statement, it would be better to highlight the blurry areas of Taming 3DGS using rendered images of fully trained models.

- From the same perspective, the claim in line 474 that "our actual rendering quality is superior to Taming 3DGS," supported by comparisons in Figure 11 at early iterations (2k, 8k), is not acceptable. This is an unfair comparison because Taming 3DGS’s slow-growing configuration clearly prevents the scene from being fully optimized at early iterations.

- In lines 214–215, the authors mention that "existing methods grow the Gaussians slowly and achieve the user-specified budget after a long-term period ... This growing strategy limits the representation power due to the limited number of Gaussians." I disagree with this statement because existing methods provide hyperparameters to control the details of the densification algorithm. For example, users can adjust the number of splits or clones, lower the positional gradient threshold, or shorten the execution frequency to accelerate Gaussian growth. Therefore, even if existing methods exhibit limited representation power, it may not be due solely to their growing strategy.

- For better readability, it would be a good idea to move some core results from the appendix into the main text, i.e., the ablation study section and Table 2.

- Insufficient ablation study leads me further wonderings:
1. What was the target number of Gaussians for the baseline?
2. The authors need to provide a more detailed breakdown to justify their modules. "Iterative Pruning" and "Gaussians Compensation" alone are not sufficient. I suggest to show the following cases:
    1. Hybrid gradient-based clone and split
    2. Position adjustment
    3. Low-quality, high-error pixel compensation
    4. Pruning

---

### Note · Authors · 2025-11-14

I have read and agree with the venue's withdrawal policy on behalf of myself and my co-authors.